# Wearable Sensors for Learning Enhancement in Higher Education

**DOI:** 10.3390/s22197633

**Published:** 2022-10-08

**Authors:** Sara Khosravi, Stuart G. Bailey, Hadi Parvizi, Rami Ghannam

**Affiliations:** 1James Watt School of Engineering, University of Glasgow, Glasgow G12 8QQ, UK; 2The Glasgow School of Art, Glasgow G3 6RQ, UK; 3Petroxin Ltd., London HA8 7JU, UK

**Keywords:** wearable sensors, engineering education, technology enhanced learning, learning technologies

## Abstract

Wearable sensors have traditionally been used to measure and monitor vital human signs for well-being and healthcare applications. However, there is a growing interest in using and deploying these technologies to facilitate teaching and learning, particularly in a higher education environment. The aim of this paper is therefore to systematically review the range of wearable devices that have been used for enhancing the teaching and delivery of engineering curricula in higher education. Moreover, we compare the advantages and disadvantages of these devices according to the location in which they are worn on the human body. According to our survey, wearable devices for enhanced learning have mainly been worn on the head (e.g., eyeglasses), wrist (e.g., watches) and chest (e.g., electrocardiogram patch). In fact, among those locations, head-worn devices enable better student engagement with the learning materials, improved student attention as well as higher spatial and visual awareness. We identify the research questions and discuss the research inclusion and exclusion criteria to present the challenges faced by researchers in implementing learning technologies for enhanced engineering education. Furthermore, we provide recommendations on using wearable devices to improve the teaching and learning of engineering courses in higher education.

## 1. Introduction

Wearable sensors are now becoming an integral part of our daily lives. Thanks to advances in technology, these devices are enabling users to seamlessly interface and interact with machines and computers. Through this interaction, users can participate in various tasks via interfaces such as the desktop computer, smartphone or any touch or gesture-based system or more advanced types of technologies such as Virtual Reality (VR) [1], and Augmented Reality (AR) [2].

In the early 21st century, a wearable device such as a wristwatch was more of an industrial design than an enabler for human–computer interaction. In fact, wearable technologies are defined as small digital devices designed to be worn on the human body [3,4]. They can incorporate wireless connectivity to access and exchange contextually relevant information seamlessly [5]. Wearable devices are increasingly prevalent in various applications including health monitoring [6,7], gesture recognition [8,9,10], entertainment [11], gaming [12], and fashion [13].

More recently, wearable devices have been used for educational purposes [14,15,16]. Recent studies report that educators use wearable sensors to improve teaching quality and Students can use these sensors to improve the interaction and engagement in classroom [17,18]. Due to their imperceptibility and direct contact with the human body, wearables can play a significant role in learning and education [19,20]. Wearable sensors functionality has improved significantly and gets more convenient, easier to interact with, and performs better in real-time [21]. Previously, a different scientific events was devoted to advances in wearable technologies for education [22]. However, majority of the reported contributors mainly focused on pre-university learner experiences. The aim of this manuscript is therefore to further review recent developments in wearable technologies for learning and to focus on their deployment in a higher education environment.

VR headsets were among the first use cases of wearable devices in education, particularly for teaching abstract subjects such as mathematics and geometry [19,23]. According to the literature, such VR technology expedited the development of immersive and collaborative learning in the classroom [24]. In addition, the extended reality (XR) and AR have been investigated widely for educational purposes [25,26,27]. Furthermore, with the expansion of wearables in education, digital augmentation of physical activities have been used for virtual field trips [28]. This work by abandoning the conventional view of IT and education, and reconceptualising information and technology in terms of “digital augmentation of physical activities”, benefits of collaborative discovery and exploration, where collecting of data and reflecting on learning was done together. In addition, other types of wearables like head-mounted display was offered to view historical events to enhance learning through experiencing and feeling history as reality [29].

The literature therefore suggests that student learning has improved as a result of using wearable devices in the classroom [30,31,32]. Benefits were demonstrated for a wide variety of subjects and age limits, from K-12 education to tertiary level higher education. Recent studies on K-12 reported that the challenges of using wearable technologies for K-12 students are health and safety and diminished perceptions of self-worth [33]. Such challenges are not considered for our study in higher education students and this work aims to present a systematic survey of wearable technologies that have been used in higher education, especially for engineering education. Based on their location, we have divided these wearables into three broad categories, which are: head-worn, wrist-worn and chest-worn, as shown in Figure 1. We will therefore discuss the merits and disadvantages of devices worn on each of these locations.

This paper is organised as follows: Section 2 provides history, market and opportunities of wearable devices in higher education in details. In contrast, Section 3 discusses the system architecture and implementation. The methodology using inclusion and exclusion criteria described in Section 4. Section 5 presents the results and discussion by summarising the advantage and disadvantage of different wearable devices in education. Section 6 draws the conclusion and future work.

## 2. Wearables in Higher Education

### 2.1. History of Wearable Technology

The first wearable was a cigarette pack sized timing device, which was hidden in shoes (Figure 2) and invented in 1955. The device was designed to predict roulette wheels in casinos and was publicly introduced in 1966 [34]. Since then, Wearable devices have evolved into various forms of accessories, watches, headsets, phones and glasses, as shown in Figure 2.

In the mid-1970s, the wearable industry and market began to grow, thanks to the introduction of calculator watches [35]. By the end of the 1970s, this market significantly expanded to the entertainment sector via the introduction of the Sony Walkman cassette player. It also expanded to the workplace via the introduction of pager devices in the 1990s. However, the boom in wearable technology only took off in 2010, thanks to the introduction of casino data bank watches [11].

### 2.2. Wearable Sensors for Psychophysiological Measures

The common psychophysiological assessment methods are:*Electrocardiography (ECG)*: ECG records the electrical signals in the heart, which are often used to measure and diagnose abnormal heart rhythm. It has been widely used in basic and clinical research. Meanwhile, electrocardiogram holds an important position in psychological research. The current psychophysiological evidence shows that heart rate is affected by external stress in most cases, so its use in the objective evaluation of psychological stress can be proved [36].*Electromyography (EMG)*: EMG measurement has been proved to be useful for studying mental load, muscle mental tension and emotions, especially facial expressions. There is a strong correlation between EMG signals and emotion changes [37]. When the mood is more pleasant, the muscles will relax, and the EMG signal will become lower. When the mood changes to an unhappy state, the muscles begin to tighten, and the myoelectric signal becomes high.*Galvanic Skin Response (GSR)*: Human organs are controlled by the sympathetic and parasympathetic nervous systems under the autonomic nervous system. However, the skin is an exception to the above statement because it is completely dominated by the sympathetic nervous system [38]. Therefore, the electrodermal activity can better reflect the psychological state of people when they are stimulated by the outside world. GSR measures by skin conductance data due to skin conductance is directly proportional to sweat secretion [39]. That makes the skin conductivity an ideal indicator to measure the activation of the sympathetic nervous system.*Electroencephalography (EEG)*: EEG is a physiological monitoring method for recording brain waves. The specific method is to use a small metal disk (electrode) attached to the scalp to detect the voltage fluctuation generated by the ion current in the brain. It measures the synchronous sum of postsynaptic potentials when pyramidal cells are excited. Recently, EEG method has been widely used in psychological research because of its unique advantages, which ensures the scientific nature and objectivity of psychological research [40].*Functional Near-infrared Spectroscopy (fNIRS)*: fNIRS is a method of optically monitoring the brain that does functional neuroimaging using near-infrared spectroscopy. It can be used to calculate the cortical hemodynamic response to brain activity. Along with EEG, fNIRS is one of the most popular non-invasive neuroimaging methods that can be applied in mobile settings. Because fNIRS has limited depth in detecting cerebral cortex [41], researchers pay more attention to the role of prefrontal cortex in emotional processing [42].

### 2.3. Market of Wearable Technology for Education

The education technology (edtech) is a multi-trillion dollar industry [43] that is growing each year. Many countries in the Organisation for Economic Co-operation and Development (OECD) are devoting more than 10% of their public spending on education [44]. The emergencies (e.g., COVID-19 pandemic over last year) have often been a catalyst for reform in many divisions, including education [45]. The COVID-19 stimulated educational organisations to implement hybrid or fully remote schooling based on emerging edtech. Such digital technologies (e.g., video conferencing tools, or online learning management software to be used on portable laptop, tablet and smartphone) usually occur within years, while due to the pandemic circumstances and physical teaching limitations, they progressed within months [46].In future, in case of any lock-downs like pandemics the edtech require to provide full remote learning and teaching.

Furthermore, public spending on edtech is projected to increase as more countries increase their public spending on education. For example, low- and middle-income countries plan to increase spending on education from the current US$1.2 trillion per year to US$3 trillion [47]. As mentioned in the Incheon Declaration, countries need to allocate at least 4% to 6% of their gross domestic product (GDP) on education; and/or allocate at least 15% to 20% of public expenditure to education [48].

In addition to this projected growth in the educational sector, the wearable technology market is growing sharply and is strongly correlated with advances in globally connected devices. These are predicted to increase from 593 million devices in 2018 to 929 million devices by 2021 [49]. In 2020, the wearables market was estimated to be worth 5 billion dollar [50].

In 2020, Vandrico INC compiled a database of companies and wearable products. According to their research, there were 266 registered companies, which have produced 431 different wearable products [51]. They also divided wearables according to 7 different categories, which were: Entertainment, Fitness, Gaming, Industrial, Lifestyle, Medical and Pets. None of these categories included education, despite the literature showing a clear use of wearables in this field. Our work therefore goes beyond the state of the art, since it aims to review the range of technologies used for educational purposes. According to their investigations, the majority of these wearables are in the lifestyle sector, with products that include SAMSUNG GEAR S3 [52], XIAOMI MI BAND 2 [53] and iHeart Internal Age fitness tracker [54]. The fitness sector is at the second place with products like Garmin ivosmart [55], Fitbit [56], Withings Hybrid Smartwatch [57].

### 2.4. Education

Creative undergraduate and postgraduate courses that rely on active learning methods are required to meet the needs of new technologies such as wearables [58,59,60]. According to Statista and the National Purchase Diary Panel Inc (NPD Group), wearable devices are popular among the younger generation, typically those aged between 18 and 39 [61,62]. Therefore, in recent years, universities have been interested in introducing wearable devices in their educational curricula [63]. Despite the rapid growth in using wearable devices in healthcare, entertainment and other applications, using wearables for education are still in their infancy.

Wearable devices can be worn on different body parts such as the head, neck, chest, torso, waist, shoulders, arm, wrist, hand, finger, legs and feet. The majority of existing devices are worn on the wrist and are mainly used for fitness purposes. Wearer comfort and familiarity could be the reason for their success. Furthermore, wearable devices such as head-mounted displays, smart glasses and smartwatches were proven beneficial for educational purposes. The locations of these wearable devices on the body is shown in Figure 1. In fact, these devices were used for different educational activities such as medical training, student engagement and authentic learning.

In the following section, we will discuss the technical architecture and specifications of wearable devices that have been exploited for educational applications.

## 3. System Operation and Implementation

Wearable devices can be divided into several main building blocks, as shown in Figure 3. These include ‘sensors’ for detecting the signals, ‘electronics’ for data processing and communications, ‘power management circuitry’ and an ‘energy harvester’. The sensors are designed according to the signal frequency and parameter range attached to adjacent human organs. They are generally placed on three sensitive body locations of head (e.g., electroencephalography: EEG), wrist (e.g., electromyography: EMG) and chest (e.g., electrocardiography: ECG), as summarised in Table 1.

Furthermore, wearable sensors should be designed to have enough sensitivity and resolution to capture the required output voltage at a specific frequency. Besides, the electronics unit plays an essential role in recording signals and cancelling unwanted noise such as motion artefacts. In addition to wearable hardware devices, software-based subsystems are needed to process and analyse the data collected by wearable sensors. The wearable hardware input and output interact with a computing device over the software interface. Wearable hardware and software need to be harmonised to enable high speed and low latency computer output.

In the following section, we will describe our approach in gathering data regarding the use of wearable devices for teaching in a higher education environment.

## 4. Methodology

This section defines our research methodology in collecting and synthesizing evidence on wearable technology application in higher education. We have demonstrated in the literature the history and evolution of wearables used in the purpose of education in universities.

First, similar to the methodology described in [64], and our previous research in [65,66], we outlined the research questions (RQs) and the inclusion criteria (InC) and exclusion criteria (ExC) of our search.

Then we will go through the advantages and disadvantages of different wearable devices used in higher education, their placement on the body, and the ideal location to be placed. The RQs are:RQ 1.What wearable used in higher education?RQ 2.Which area on the body is best for placement of wearable?

The InCs are:InC 1. Wearable devices used for teaching and learning in any discipline.InC 2.The higher education level of study (undergraduate).InC 3.Only include programs conducted in English.

The ExCs are:ExC 1.Smartphone as a type of wearable device.ExC 2.Wearable in medical purposes.ExC 3.Professional certificates or extra-curricular activities.

To collect research papers that match our criteria, we have used Web of Science and Google Scholar for surveying the literature. We have used the descriptors and synonyms summarized in Table 2 for our search. Four considered descriptors include “wearable”, “higher education”, “undergraduate” and “engineering”.

## 5. Results and Discussion

As previously mentioned, there has been a steady growth in the number of publications related to wearable devices. A similar trend is apparent for the number of publications related to wearable devices and education, as evidenced by Figure 4, which shows the number of research publications related to wearables. Clearly, the literature shows an increased interest in wearable devices since the 1970s. A comparable but delayed trend can be noticed with the number of publications on wearables in education, since academic interest in this area only began in 1994. From Figure 4, it is noteworthy to mention that interest in wearables for education follows a similar trend to wearables, which is scaled down by a factor of approximately 143.

However, a total of 20 studies matched our InC and ExC criteria, which were defined in the Section 4. As previously mentioned in the Introduction, we have classified wearable devices for higher education according to three categories: head-worn, wrist-worn and chest-worn. A summary of each of these technologies is presented in Table 3. In this section, we discuss the wearable technologies that have been implemented in each category for higher educational purposes.

### 5.1. Head-Worn Devices

Head-worn devices or displays have come a long way since the early 1970s [87,88]. In 1968, Ivan Sutherland demonstrated the first head-mounted display at Harvard University [87]. During the past four decades, scientists and researchers have been investigating ways of developing full colour opaque displays that enable users to see through to the real-world. However, thanks to advances in the microelectronics industry and the emergence of Light Emitting Diodes (LEDs), this is now a reality. Examples of commercial devices include Microsoft’s Hololens [89]. In addition to the range of wearable sensors that currently exist for the learning enhancement, wearable eye trackers, enabling them to record subtle eye movements in different head movements and directions [90,91].

In terms of their use in higher education, the University of South Australia found evidence to support the impact of real-time information overlay on learning using head-mounted displays [67]. EEG sensors were used in educational design programs, to assess brain activity through the wearable plug and play headset, combined with Oculus Rifts VR to conduct spatial assessments [68]. Emotiv EPOC^®^ EEG head-mounted gaming system Figure 5A has been used in cognitive and brain science to measure brain activity in the Macquire University [69]. In [70], they investigated the use of VR on the performance of computer engineering bachelor of science students.

In addition to the head-mounted displays, glasses (e.g., Google Glass shown in Figure 5B) have been used in many different educational purposes such as surgical education [71,72,92] and in some cases during the encounter with standardized patients to record their first-person perspectives [73,74]. In Ohio State University, the Google Glass was used by a surgeon to broadcast a surgery live to a group of medical students [75]. Another study used Google glasses in educational psychology and organizational behaviour classrooms [76]. In another study, users at the University of Illinois, Chicago, interacted with complex 3D objects in Cave Automatic Virtual Environment via special glasses to observe the object of study from diverse angles [77]. Epson smart Glass was used in this case for environmental education in Murdoch University [78], and in another study, they used Epson glasses for language learning purposes [79]. Google glasses were also used to provide teaching performance feedback to teachers and to improve social and communication skills in students, and teacher relationship [80].

Table 4 summarises the main advantages and disadvantages of head-worn devices. We discuss these in the following sections.

#### 5.1.1. Advantages of Head-Worn Wearables

Head-worn wearable devices such as head-mounted displays (HMD) offer alternate learning styles with the use of VR, for students who are mainly visual, active, and global learner [98,99]. The use of before-mentioned wearables has been grown by advancing in technical specifications. For instance, the significant changes in the application of HMD happened in 2016, when their ‘field of view’ expanded from 25 and 60 degrees to above 100 degrees [100]. This new feature gives the first-person experience and sense of presence to the user. It makes it possible to experience, situations that are either inaccessible or problematic, e.g., in space studies [94,101].

#### 5.1.2. Disadvantages of Head-Worn Wearables

On a different note, because of web access and the collection of personal data, privacy is of significant concern regarding wearables such as HMD and glasses [76,92]. In addition to privacy and security, the head-worn wearable devices can be complicated to use for individuals with no prior knowledge. Therefore, this complexity requires establishing a professional training session for both learner and educator [94]. Another disadvantage of head-worn wearable devices is related to their high cost. For example, producing a VR simulation is expensive and educators will usually use VR simulators available in the market that are not explicitly made for that content [94]. These simulations are not usually specialised for teaching deployment in a classroom but can be used for self-learners. As the last drawback, head-worn wearable devices can cause cybersickness, because of the extended amount of time used [94].

### 5.2. Wrist-Worn

Devices worn on the wrist can be traced back to the beginning of the 20th Century, when Louis Cartier invented the wristwatch for the Brazlian aviator Alberto Santos-Dumont’s [102]. Currently, due to their unobstructiveness to the wearer, various types of wearable devices have been developed, which have also been used for educational purposes. Examples include:Wristbands: wearable wristband include sensors that can collect bio-signals that can be used to estimate stress in students [81]Watches: Smartwatches can facilitate surgical training to improve clinical education by exploiting motion-based metrics and offering a new source of feedback through objective assessment Figure 5C [82].ECG Sensors: At Imperial College London, students from the Advanced Signal Processing and Adaptive Signal Processing and Machine Intelligence courses used a custom-made wearable ECG recording device to measure the level of student engagement, and learning [83].

In [84], wrist-worn activity trackers equipped with bio-metric sensors were used in higher education regarding eHealth literacy acquisition.

#### 5.2.1. Advantages of Wrist-Worn Wearables in Education

Wearable devices like wristbands can collect physiological signals of the human body and are proven beneficial for education purposes. For example, wristbands used to monitor the high-stress level caused by burnout syndrome, which is a common factor in students at universities and can result in a higher number of dropouts [103]. Because of their popularity and situated place on the body, wristbands can be used with a higher number of students in a group assessment. Furthermore, it offers the user the ability to move around freely without interrupting the result. Such movement ability is limited in case of head-worn and chest-worn [81]. In another study, wrist-worn wearables were employed to replace assessment tools used to measure trainees performance as these traditional tools are not always sensitive enough to detect different levels of expertise [82]. Furthermore, students can use wrist-worn wearables to collect their own data. This feature gives the sense of engagement to students. The data collected from wrist-worn patches like ECG signals have significant value both in clinical and non-clinical applications because of the ease of interpretation, reliability, and physiological meaningfulness [83].

#### 5.2.2. Disadvantages of Wrist-Worn Wearables in Education

While wrist-worn wearables are quite popular, they might experience some drawbacks. For example, the wireless connection between wearable and smartphone may be broken during the test, and this could result in inaccurate data [81]. Moreover, if the wearable is placed on the forearm, it reduces the ability to move around quickly [83]. Whether it is worn tight or loose, the distance between sensor and skin can change the collection of data. Another shortcoming is affected by the wrist-worn battery life; depending on the functions used, the battery can require recharging, or exchanged, regularly.

### 5.3. Chest-Worn

Examples of chest-worn devices used for educational purposes include the wearable chest plate. In a study, students and faculty from engineering and nursing developed a wearable Tracheostomy Overlay System (TOS), shown in Figure 5D, for use with standardized patients. This device was designed to improve the education of health professional students while learning assessment and care of a patient with the tracheostomy in clinical practice [85]. Another study used sociometric badges with wearable sensors to collect social interaction data for predicting collaboration quality and creative fluency outcomes [86].

#### 5.3.1. Advantage of Chest-Worn Wearables in Education

Chest-worn wearables are embedded with different sensors that can collect and store data such as heart rate, heart rate variability, and respiration. Sociometric sensors are one type of chest-worn wearable that are applied to collect social interaction data. The sociometric badges combine sensor technologies including Bluetooth and infrared sensors, an accelerometer, and microphones, to capture several variables about speech and conversation dynamics, body movement and posture, and social proximity [86]. Chest-worn wearables were also employed to track stress within nursing students that enabled assessment of physiological changes and the collection of subjective responses to the origin of stress [96].

#### 5.3.2. Disadvantages Chest-Worn Wearables in Education

Chest-worn wearables, such as social interaction sensors, can collect a comprehensive picture of users’ social networks and performance. Approval of these sensors extensively depends upon assuring users’ privacy as the primary concern.

A summary of our findings on wearables for higher education purposes is presented in Table 4.

## 6. Recommendations

There is a need to explore physical education educators’ perspectives and the universities on their readiness to deploy and integrate wearable sensors as an innovation in physical education and to develop a conceptual model for integrating wearable senors [46]. As demonstrated from our literature review, there is a strong interest in using wearable devices to improve the teaching and learning of engineering courses in higher education. Curriculum developers have been experimenting with a range of devices on three main body parts: the wrist, head and chest. In comparison to the healthcare industry, their integration in engineering curricula has not been validated, nor standardised, and there is still evidence to suggest that the application of such technologies improves student satisfaction or performance. Therefore, an implementation route, or validation process is required to ensure that curriculum developers fully exploit the benefits of integrating wearable devices in a higher education setting.

As previously mentioned, such validation processes have been reported for wearable devices used in the healthcare industry [104]. Therefore, for the full adoption of wearable devices, there is a need for developing comprehensive guidelines to standardise their use in a higher education environment. Here, as shown in Figure 6, we recommend a three-step acceptability route that includes (1) content validation, (2) feasibility and features, and (3) implementation.

We summarise these steps as follows:

### 6.1. Content Validation

This step includes problem analysis, which is utterly necessary in order to identify and understand the teaching and learning issue. The factors and elements that affect the process of student learning need to be considered. Furthermore, this step assesses both students and instructors needs via a process of identifying and defining them. Such needs assessments are important so that a careful state-of-the-art review in wearable technology is evaluated [105]. Through this literature review, different types of wearable devices can be identified and their risks can be assessed. The risk assessment involves defining and analysing potential events that may negatively affect students, instructors, and the overall learning environment. Carefully constructed surveys and questionnaires can be used to identify general opinions in this step.

### 6.2. Feasibility and Features Study

Data security is the first item to be considered in this step. For example, it is important to protect both student and student’s personal data from unauthorized access, since this is among the major concerns of using wearable devices in higher education [93]. In addition, the physical size and mobility features of wearable devices to ensure maximum comfort and wearability need to be considered for an extended time. Cost effectiveness should also be considered for comparisons with traditional learning that do not involve wearables. Cost might also be important with respect to digital poverty. A situation that has been highlighted and exacerbated by COVID-19 and online learning, where not all students have access to the digital technology required to have equal access to the learning. Furthermore, long battery-life and large data storage specifications of wearable devices improve the feasibility of their educational deployment. Short-battery life and inconvenience of recharging is time-consuming, and insufficient storage capacity for collecting a continuous stream of data cause inaccuracy in the learning and training scheme. The last term to accomplish this step is adaptability of wearable devices to suit students, lectures and different settings [106].

### 6.3. Implementation

This step covers the standardisation of wearable devices to attain the certainty that processes associated with their creation and performance are delivered within set guidelines. In this regard, expert opinions will be collected to get a better quality end product. For example, external examiners who are experts in wearable devices might be needed. Afterwards, will be evaluated against a gold standard; an advanced standardised wearable device in each classification of wearables that the rest of developing devices can be compared with [104,107]. Lecturers and instructors, who are considered the primary source of knowledge need to be familiarised with the wearable devices and their affordances before introducing them to students. They also need to be trained and familiarised with inclusive and active teaching approaches that effectively engage the learner [108,109]. The last item in the implementation step to be executed is rules and regulations that are systematically arranged around the production and application of wearable devices [110].

## 7. Conclusions

With daily advances in engineering research, there is potential for these novel wearable devices to be implemented for educational application, enhance the students learning and schooling skills, and provide better alternatives for the future. This study investigated the advantages and disadvantages of wearable technologies in higher education. We presented the challenges associated with employing wearable devices in engineering education to enhance learning. We have recommended an acceptability route to implement wearable devices in the higher education environment. We believe that introducing wearables into the classroom is now feasible, and a game-changer technology in engineering education. Expectedly, our data collections motivated by this study will suggest additional investigative methods of supplementing the wearable technology curriculum with appropriate real-world cases.

## Figures and Tables

**Figure 1 sensors-22-07633-f001:**
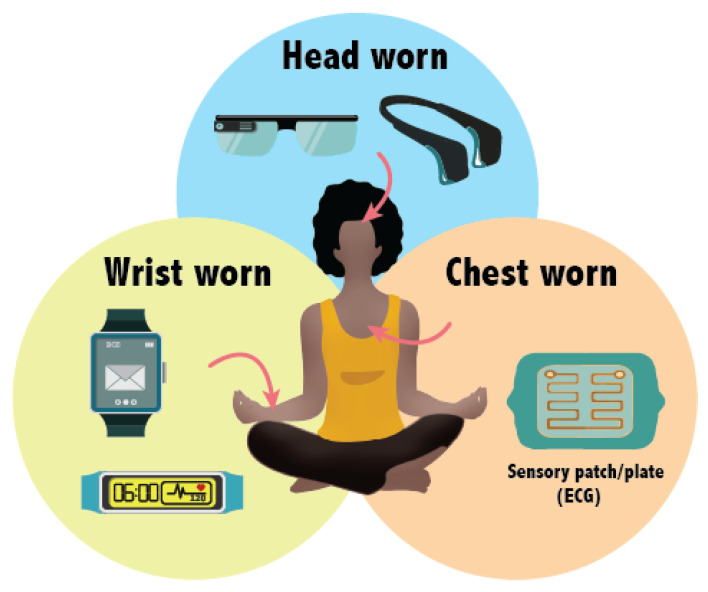
Classified wearable sensors for educational purposes according to three major categories according to their placement. They can be worn on the head, wrist or chest to collect and monitor information from students and teachers.

**Figure 2 sensors-22-07633-f002:**
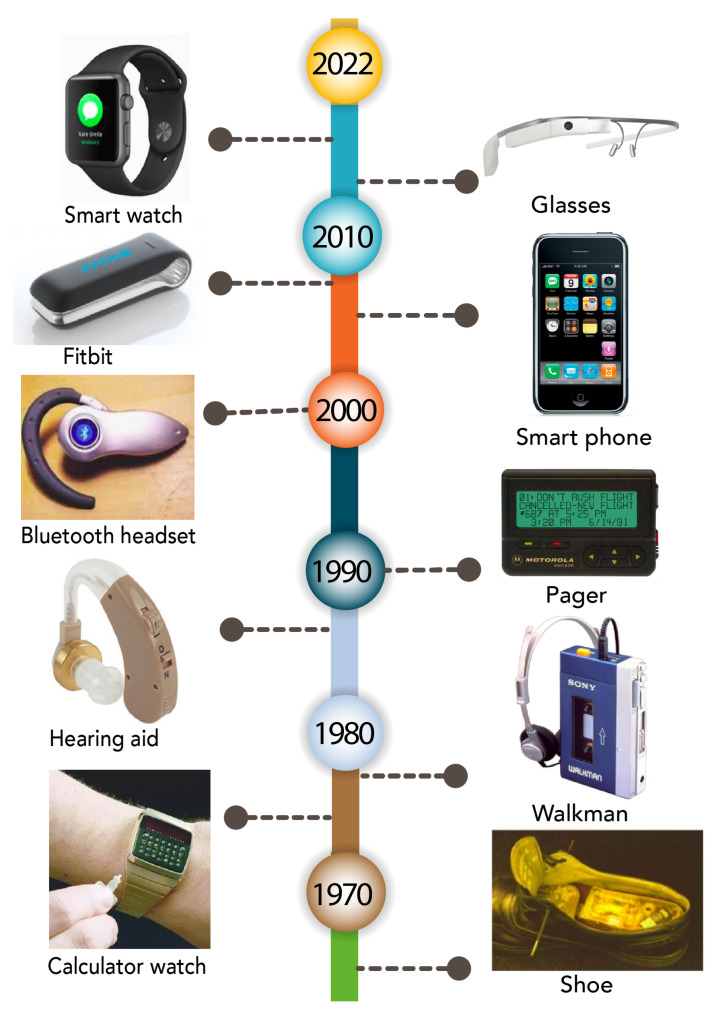
Development of wearable devices during the past 50 years. Starting from 1960s, the first wearable product was a centimeter-scale computer hidden inside shoes. Currently, advanced millimeter-scale systems are embedded on wrist-worn, chest-worn and head-worn platforms.

**Figure 3 sensors-22-07633-f003:**
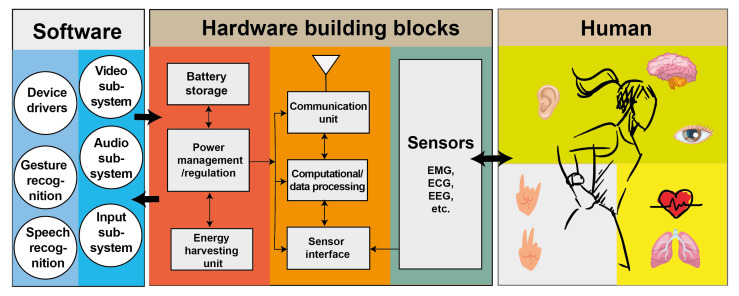
Concept diagram showing the software and hardware building blocks of wearable devices for educational purposes. The main hardware building blocks of a wearable device are the sensors, readout circuit interface, the energy harvester as well as the power management and telecommunications units. The software component can be programmed to drive the wearable’s hardware according to different subsystems and inputs from sensors, e.g., video, audio, gesture and speech.

**Figure 4 sensors-22-07633-f004:**
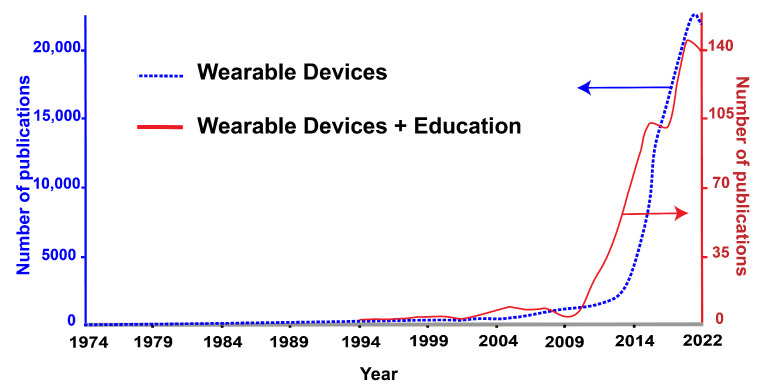
Plots showing research publications in the field of wearable devices (blue *y*-axis on the left side), and wearable in education (red *y*-axis on right side) since 1974. The data were extracted from Web of Science by searching keywords such as “wearable devices”, “wearable” and “education”.

**Figure 5 sensors-22-07633-f005:**
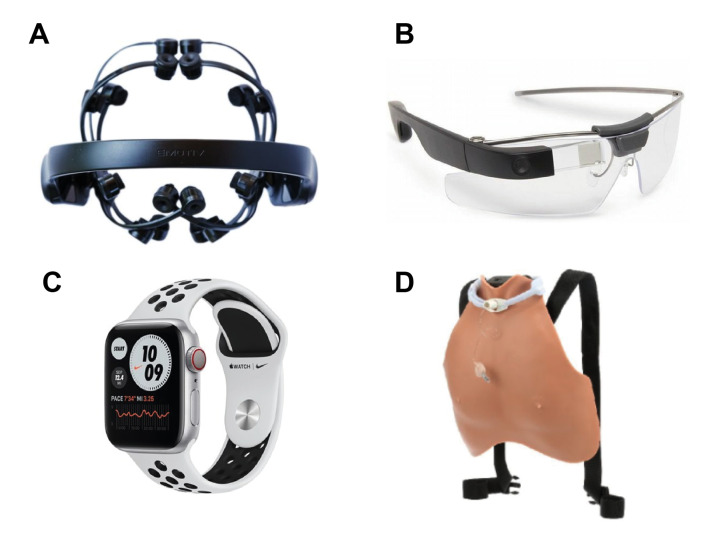
Various head-worn wearable have been used for educational purpose including (**A**) Emotiv EPOC EEG system [69] (**B**) Google Glasses in [71] and (**C**) Wearable wrist-worn like Apple Watch used in [82] and (**D**) Wearable chest-worn studied in [85].

**Figure 6 sensors-22-07633-f006:**
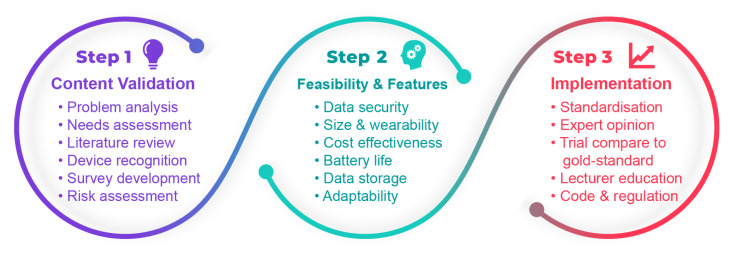
Acceptability route to implementing wearable devices in a higher educational setting.

**Table 1 sensors-22-07633-t001:** Technical specifications for various wearable sensors to collect the necessary physiological biosignals required in educational studies, namely, ECG and heart rate variability, EMG and EEG.

Sensor Type	Signal Frequency	Parameter Range
Chest-worn, e.g., ECG sensor	250 Hz	0.5–4 mV
Wrist-worn, e.g., EMG sensor	10–5000 Hz	0.01–15 mV
Head-worn, e.g., EEG sensor	0.5–60 Hz	0.0003 mV

**Table 2 sensors-22-07633-t002:** Descriptors and synonyms.

Descriptor	Definition	Synonyms
Wearable Technology	This is a category of electronic devices that can be worn as accessories, embedded in clothing, or even tattooed on the skin.	Body attached technology
Higher Education	Refers to a level of education following secondary or high school. It takes places at universities and Further Education colleges and includes undergraduate and postgraduate study.	Tertiary education
Undergraduate	Refers to education conducted after school and prior to postgraduate education and includes all post-secondary programs up to the level of a bachelor’s degree.	Bachelor’s degree

**Table 3 sensors-22-07633-t003:** Wearable devices in higher education and main features.

Main Category	Sub Category	Application Targets	References
Head-worn	Head-mounted and Glasses	EEG, cognitive and brain science, surgical training, simulation-based training atmospheric scientists or detail hurricanes, environmental education	[67,68,69,70,71,72,73,74,75,76,77,78,79,80]
Wrist-worn	Watches and Wristband	Estimate stress in students, motion-based metrics to improve clinical education, ECG signal	[81,82,83,84]
Chest-worn	Patch sensors	Occupational stress, collaboration quality and creative fluency	[85,86]

**Table 4 sensors-22-07633-t004:** Advantages and disadvantages of wearable devices in higher education.

Categories	Advantages	Disadvantages
Head-worn	First person point of view [73,75], access to difficult and impossible places [93], seamless and fast access to information [76], spatial and visual awareness [94], students feeling a deeper connection with learning materials, deeper student analysis and understanding of scenario-based practices [69,93], record and retrace interpersonal communication skills and nonverbal behaviours [73] video recording [72]	Cyber sickness [94], lack of content [94], technical limitation [94], privacy concern [76,95], connectivity issues [76], hardware failure [73], physical discomfort [79].
Wrist-worn	Data collection from large group of students [81], automatic data collection [81], low maintenance [81], no disruption to classroom [81], increases student engagement by collecting their own physiological data [83], easy functionality easy to interpret [83]	Disconnection between wristband and secondary device [81], hardware-related issues such as compromised sensor sensitivity, battery life [81] and wearer movement [83].
Chest-worn	Collect data automatically and without interruption [86], for collection of social interactions data [86], continuous record heart-rate, heart-rate variability, respiration, and physical activity [96].	Lack of user privacy [97].

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
