# Peer review of "Wearable Sensors for Learning Enhancement in Higher Education"

_sensors, 2022, doi:10.3390/s22197633_

Round 1
Reviewer 1 Report
Wearable sensors are being applied in education more and more, especially after the emergence of the COVID-19. Therefore, discussing the wearables used in higher education is meaningful and helpful. The paper gives people a systematic view of wearables used in higher education. There is one more thing need to be clarified, what is the difference between the wearable sensors used in the higher education with that in K-12 and others.
Author Response
We thank the reviewer for this valuable comment. Indeed, wearable devices have been used in other educational sectors, and we highlighted the importance of wearables for higher education purposes, especially in engineering degrees in universities. In this regard, we clarified this note in the Methodology (Section 4) that our study is for the higher education level of study (undergraduate). Recent studies on K-12 reported that the challenges of using wearable technologies for K-12 students are health and safety and diminished perceptions of self-worth [Ref 1]. However, such challenges are not considered for our study in higher education students. According to our search criteria, our inclusion criteria are presented for the first time, which is essential for future deployment of wearables to enhance learning and training in universities. We have revised the manuscript according to the reviewer's comment.
[Ref 1] Jovanovic, P. and Kay, R., 2021. Examining the Use of Wearable Technologies for K-12 Students: A Systematic Review of the Literature. Journal of Digital Life and Learning, 1(1), pp.56-67.
Reviewer 2 Report
Reviewer: 1
Comments to the Author:
In this manuscript, the authors presented an overview of wearable devices that have been utilized for teaching and delivering engineering curricula in higher education. They also summarized the advantages and disadvantages of these devices based on the locations in which they are worn on the human body. This work is very useful for the community and offers new references for scholars. However, I have several minor concerns before this manuscript can be accepted. Therefore, in its current form, revisions are needed.
1. To collect the necessary physiological biosignals required in educational studies, namely, ECG and heart rate variability, EMG and EEG. The developments should be elaborated comprehensively.
For example, Rogers et al developed a biosoresorbable silicon electronic sensors for the brain. (10.1038/nature16492).
Author Response
We thank the reviewer for this valuable comment. The reviewer is suggesting a discussion around different types of sensors. We have revised the manuscript according to the reviewer's comment and elaborated on the development of various sensors that can be used for collecting physiological data from participants in an educational setting. However, we did not include the paper by Rogers et al since it is not in the teaching and learning domain. Their focus is on the medical field.